# Reduction of Alcohol-Dependent Lung Pathological Features in Rats Treated with Fenofibrate

**DOI:** 10.3390/ijms252312814

**Published:** 2024-11-28

**Authors:** Diego A. Rojas, Krishna Coronado, Diliana Pérez-Reytor, Eduardo Karahanian

**Affiliations:** 1Instituto de Ciencias Biomédicas (ICB), Facultad de Ciencias de la Salud, Universidad Autónoma de Chile, Santiago 8910132, Chile; krishna.coronado@cloud.uatonoma.cl (K.C.); d.perez@uautonoma.cl (D.P.-R.); 2Research Center for the Development of Novel Therapeutic Alternatives for Alcohol Use Disorders, Santiago 8910132, Chile

**Keywords:** fenofibrate, inflammation, ethanol, hypersecretion, fibrosis, airways

## Abstract

Alcohol use disorder (AUD) is a public health problem characterized by a marked increment in systemic inflammation. In the last few years, it has been described as the role of alcohol in neuroinflammation affecting some aspects of neuronal function. Interestingly, inflammation is reduced with fenofibrate treatment, a PPARα agonist used to treat dyslipidemia. On the other hand, alcohol has been associated with chronic inflammation and fibrosis in the lungs, affecting their normal function and increasing respiratory infections. However, a deep characterization of the role of alcohol in the worsening of chronic respiratory diseases has not been described completely. In this work, we present a novel study using rats treated with alcohol and fenofibrate to evaluate the relevant features of chronic respiratory disease: inflammation, mucus hypersecretion, and fibrosis. The analysis of extracted lungs showed an increment in the inflammatory infiltrates and pro-inflammatory cytokine levels associated with alcohol. Interestingly, the treatment with fenofibrate decreased the expression of these markers and the infiltrates observed in the lungs. The levels of mucin Muc5ac showed an increment in animals treated with alcohol. However, this increment was markedly reduced if animals were subsequently treated with fenofibrate. Finally, we documented an increment of collagen deposition around airways in the animals treated with alcohol compared with control animals. However, fenofibrate treatment reduced this deposition to a level similar to the control animals. These results showed the role of alcohol in the increment of pathological features in the lungs. Moreover, these features were attenuated due to the fibrate treatment, which allows us to glimpse this drug’s promising role as lung anti-inflammatory therapy.

## 1. Introduction

Alcohol use disorder (AUD) has been widely described by the World Health Organization (WHO) as a health and social concern (https://www.who.int/news-room/fact-sheets/detail/alcohol, accessed on 21 Ocober 2024). Sadly, despite the existence of pharmacological treatments to reduce alcohol consumption in AUD patients, such as naltrexone and acamprosate, most of the patients relapse after the initial detoxification treatment [1,2] and stop taking the medication or receiving an appropriate treatment [3]. These points are critical to the development of an effective drug for the treatment of alcohol addiction. In addition, there is evidence that links AUD with various health problems, including cardiovascular diseases [4,5], liver disease [6], and central nervous system diseases [7,8]. Interestingly, one of the critical factors in the increasing risk of these diseases is the induction of inflammation-related pathways observed in the gut, liver, and brain [9,10,11]. Moreover, alcohol consumption has been linked to a loss of cognitive function [7,12] and an accelerated process of brain aging [13,14]. The inflammation observed in the brain may be related to ethanol oxidation and the increment of reactive oxygen species (ROS), leading to the induction of the NF-kB pro-inflammatory pathway [15,16]. Additionally, NADPH oxidase (NOX) is activated under this mechanism, leading to a more significant increase in the release of ROS [17]. Another mechanism associated with inflammation and alcohol consumption is a pathway activated by the rise of intestinal mucosa permeability, allowing the entrance of bacterial lipopolysaccharide (LPS) to the blood, inducing tumor necrosis factor-alpha (TNFα) release that initiates a systemic inflammation [18,19]. Independent of which mechanism is activated by alcohol consumption, both release pro-inflammatory cytokines such as TNFα, IL-1β, and IL-6 [18]. Recently, a promising treatment to reduce neuroinflammation has been described in alcohol-drinking rats treated with fenofibrate. The treatment with fenofibrate, a PPARα antagonist, leads to a decrease in neuroinflammation and a reduction in alcohol relapse [20,21]. The mechanism by which fenofibrate decreases neuroinflammation is given by the ability of PPARα to inhibit the activity of NF-κB rather than diminishing its expression [22]. Specifically, the administration of fenofibrate has been found to inhibit TNFα-induced IκB kinase (IKK) activity and, therefore, IκB phosphorylation [23]. In this way, NF-κB remains sequestered in the cytoplasm, and as it cannot be translocated to the nucleus, it cannot activate the transcription of its target genes.

Furthermore, the lungs are particularly susceptible to alcohol intake due to the vaporization of ethanol in airways, leading to chronic exposure of the airway epithelial cells to alcohol [24]. It has been described that alcohol intake is associated with an 8-fold increment in the risk of pneumonia [25] and with the increment of the severity of the symptoms in several respiratory diseases, such as acute respiratory distress syndrome (ARDS) and chronic obstructive pulmonary disease (COPD) [26,27]. Alcohol directly affects lung function in several ways, for example, impairing mucociliary clearance, a mechanism that protects the airways from inhaled pollutants and pathogens. Alcohol may stimulate the ciliary beat frequency by stimulating the endothelial nitric oxide synthase pathway [28], modulating the clearance of pathogens in the airways [29]. Moreover, chronic alcohol exposure may lead to a desensitization of the ciliary response in the airway epithelium, which is described as an alcohol-induced ciliary dysfunction (AICD) disorder in individuals [30]. Another event associated with alcohol exposure is the impairment of the innate immune system, characterized by the exposure of alveolar macrophages (AM) to ROS, mainly produced by the increment of NADPH oxidase activity [31] and a decrease in the expression of anti-oxidant genes [32], leading to the impairment of the innate immune response, for example, against pathogens. An increase in oxidative stress is potentiated by reducing anti-oxidant pathways such as Nrf2, which are associated with lung injuries [33]. The adaptative immune system is also affected by chronic exposure to alcohol, affecting the standard response of several cytokines, such as interferon-γ (IFNγ) [34] or IL-17 [35], leading to a decrease in the recruitment of immune cells such as neutrophils or lymphocytes. In addition, it has been shown that alcohol increases the permeability of the alveolar–capillary barrier, allowing the infiltration of immune cells and the release of pro-inflammatory cytokines [36].

The evidence above strongly supports the hypothesis that alcohol intake leads to lung inflammation. These results highlight the need for effective therapies to reduce the inflammatory effects of alcohol in patients with AUD. However, the role of the drug fenofibrate in decreasing ethanol-induced lung inflammation and damage is entirely unknown. Interestingly, the anti-inflammatory effect of fenofibrate has been reported in another mouse model of lung inflammation and allergic asthma [37]. In this study, 30 mg/kg/day of fenofibrate administration reduced NF-kB activation in the lungs of mice previously induced with asthma.

In this work, we document that the treatment with fenofibrate reduces ethanol-induced lung lesions associated with lung diseases such as inflammatory infiltrates, mucus hypersecretion, and collagen accumulation around airways. Our results show that fenofibrate may be helpful and explored for further research as an anti-inflammatory drug in treating AUD or other respiratory diseases where chronic inflammation is a crucial factor, such as COPD or asthma.

## 2. Results

### 2.1. Fenofibrate Reverts Lung Inflammation Induced by Ethanol

As described previously, lung inflammation is one of the main features of several pulmonary diseases and may be induced by ethanol intake. To measure the inflammation status in the experimental animals described in Figure 1 and Section 4.2, lung sections stained with hematoxylin and eosin (H/E) were analyzed using 100× magnification.

To this end, we used a semi-quantitative score approach based on the percentage of infiltrated area of each section, as described in Section 4.3. We found that the infiltrated area of the lung sections was incremented in the animals injected with ethanol compared with the control animals (Figure 2). In addition, the control animals treated only with fenofibrate show the same level of infiltration as the control animals. Interestingly, animals injected with ethanol and then treated with fenofibrate showed a low infiltrated area in the same level of control and fenofibrate groups (Figure 2). The infiltration observed in the lungs of animals injected with ethanol may be due to the increment of pro-inflammatory cytokines secreted in the organism. Therefore, it is essential to determine some of the pro-inflammatory markers associated with the intake of ethanol previously described to relate to our histological findings. In previous studies, cytokines TNFα and IL-6 showed increased levels in the brains of animals stimulated with ethanol [21]. With this information, we evaluated the mRNA levels of these cytokines in the lungs of the experimental animals by RT-qPCR. Both cytokines, TNFα and IL6, showed a marked increment in the ethanol group compared with the other experimental groups, including the animals injected with ethanol and treated with fenofibrate (Figure 3). Overall, these results showed that the treatment with fenofibrate may decrease the inflammatory infiltration and the pro-inflammatory markers TNFα and IL6 in the lungs of the animals injected with ethanol.

### 2.2. Fenofibrate Reverts Mucus Hypersecretion in the Airways Induced by Ethanol

Alcohol induces the increment of the severity of chronic respiratory diseases such as COPD and ARDS, which are characterized by chronic inflammation and mucus hypersecretion. However, there is not enough evidence showing the relationship between alcohol intake and mucus in the airways. Still, principally, this relationship is associated with the mucociliary function of the airway epithelium [25]. Due to our previous experience evaluating mucus hypersecretion in the airways, we investigated the putative changes in mucus secretion in our experimental animals. To identify changes in the mucus levels in the airways, we stained lung slices from animals of all the experimental groups. We used the specific staining alcian blue (AB) against mucus proteins, as described in Section 2. The animals injected with ethanol showed an increment in the stained area of the airway epithelium compared with the control animals (Figure 4A,B). Remarkably, this increment was reduced in the animals injected with ethanol and treated with fenofibrate in a significative manner (Figure 4A,B).

These increments in the mucus present in the airways of the animals injected with ethanol may be related to changes in the expression of mucus-related genes, like the main protein component of the mucus in the lung: MUC5AC. To evaluate the levels of MUC5AC, protein extracts from the lungs of the animals were assessed by Western blot. MUC5AC protein levels were incremented in the animals injected with ethanol compared with the controls (Figure 4C,D). However, these increments in the protein levels of MUC5AC were decreased in the animals injected with ethanol and treated with fenofibrate to the same levels as the controls (Figure 4C,D). These results agree with the detection of mucus in the airways measured with AB staining. Overall, the changes in the levels of MUC5AC indicate that ethanol may be a hypersecretion inductor in the airways. Remarkably, fenofibrate decreased the mucus increment in the airways of animals injected with ethanol.

### 2.3. Fenofibrate Reverts Collagen Deposition Around the Airways Induced by Ethanol

Fibrosis of the airways is a feature described in several chronic respiratory diseases such as COPD and asthma. Moreover, some work documents that chronic ethanol exposure may increase lung collagen deposition [38,39,40]. To investigate the effect of ethanol injection in the lungs of our experimental animals, we evaluate the presence of collagen around the airways using a histological approach, as described in Section 2. The collagen levels deposited in the airways of animals injected with ethanol were the highest compared with the control animals (Figure 5). Furthermore, we evaluated the effect of fenofibrate on collagen deposition. To this end, we measure the collagen deposited around the airways of the animals injected with ethanol and treated with fenofibrate. Interestingly, the collagen levels measured in the airways were comparable to those observed in the control animals (Figure 5). These results indicate that fenofibrate could positively regulate the matrix remodeling induced by ethanol on airways.

## 3. Discussion

In this study, chronic exposure to ethanol resulted in effective lung damage, as previously described by several works [25,27,41]. We observed impairment of mucociliary clearance in the airways, impairment of immunity, alteration of epithelial permeability, and induction of matrix remodeling. The main explanation for this damage is the systemic effect of ethanol metabolization, leading to ROS release and an increment of toxicity [29]. However, the complete mechanisms associated with these alterations are not entirely understood. In our work, the first main result was the visualization of a significant increment of inflammatory infiltrates in the lungs of the animals treated with alcohol compared with the control animals. This increase was entirely reversed by treatment with fenofibrate. In addition, we aim to study the expression of two of the most significant pro-inflammatory cytokines: IL6 and TNFα. These results are consistent with previous reports in animal models of AUD, where the levels of inflammatory markers were increased [42]. IL6 and TNFα have been described as part of the inflammatory response of alcohol in neuroinflammation [21] and as part of the inflammation of the airways exposed to alcohol [43].

In our work, the effect of alcohol on mucus secretion was significant, indicated by the experiments documented by histological and molecular approaches. The evidence showed a slight relation between mucus and alcohol, where only the mucociliary system was affected. This could explain the increase in respiratory infections that lead to diseases. In our case, we documented a marked increment of mucus secretion in the airways of the animals treated with alcohol. This point is attractive due to the poor previous evidence of mucus increment and ethanol, where the interest only focused on the mucociliary system’s function. The poor evidence relating alcohol and mucus secretion in airways is only supported by studies measuring the increment in mucin expression dependent on time and the alcohol concentration in the trachea and bronchus [44].

Concerning our results, the same inflammatory effect observed in the lungs of the animals exposed to ethanol probably induced the increment of mucus. It has been described previously that the increment of inflammation via the Th2 immune response may increase mucus release in the airways, especially following STAT6 pathway stimulation [45,46,47,48]. Furthermore, alcohol intake leads to an increment in oxidative stress, leading to the increment of several pathways, including EGFR [49]. It has been described that activation of the EGFR pathway induces mucin expression in several models [50,51,52]. These points are relevant because further studies may focus on evaluating the status of the STAT6 and EGFR pathways in the lungs during alcohol intake.

In this work, we showed that alcohol treatment may induce an increment of collagen deposition around the airways. This evidence is related to previous works linking alcohol and lung fibrosis [40,53,54]. One of the mechanisms associated with this point is the pro-fibrotic pathway TGF-β1 [53] induction. Furthermore, it is necessary to evaluate the status of this pathway in the lungs of the animals exposed to ethanol to identify a putative mechanism associated with our results.

The use of fenofibrate as an anti-inflammatory treatment in previous studies using animal models of alcoholism showed a promising role in a decrease in inflammation and oxidative stress markers in the brain [20,21]. This led us to the evaluation of fenofibrate for the treatment of the pulmonary disease features induced in our animal model with alcohol injections. At this point, we identify a marked decrease in inflammation infiltrates, a decrease in pro-inflammatory cytokines IL-6 and TNFα, a decrease in mucus detection in the airways, and a decrease in collagen deposition. These results show the potential role of fenofibrate as a new inflammatory modulator for reversing the inflammatory effects of alcohol in the lungs. Interestingly, fenofibrate activates the PPARα pathway, negatively modulating a specific pro-inflammatory pathway, NF-kB, as described in neuroinflammation [20]. This feature would mark a distinction between fenofibrate with respect to the classic non-steroidal anti-inflammatory drugs (NSAIDs) since the former would inhibit the innate and adaptive immune responses at the level of its master activator (NF-kB). In contrast, NSAIDs act only at the level of cyclooxygenases, inhibiting prostaglandin production. However, in our case, we describe an effect on mucus secretion related to several mucogenic pathways, such as STAT6 or EGFR, and an impact on collagen deposition. We believe that the inhibition of mucus secretion may be associated with an overall decreased inflammation. The association between the NF-kB pathway and mucus hypersecretion has been described previously [55,56,57], so this connection is probable in our model.

On the other hand, the relation between fenofibrate and collagen deposition is unknown. A putative explanation is the relation between pro-collagen pathways and other inflammatory pathways associated with NF-kB, where inhibiting these pathways decreases collagen deposition and lung injury [58,59,60]. In addition, we still do not know if the evidence points to a reversion of the collagen deposition in the resting time after ethanol application and during the fenofibrate treatment or if the role of fenofibrate is to inhibit the increasing collagen deposition. To this end, it is necessary to conduct additional experiments following the changes in the airways in a time-dependent manner.

In conclusion, we have described the promising role of fenofibrate in decreasing the standard pathological features identified in chronic respiratory diseases such as asthma and COPD. Further experiments must be performed to define the mechanisms associated with the role of fenofibrate in these features identified in the lungs of animals treated with alcohol.

## 4. Materials and Methods

### 4.1. Ethics

Animal experiments were performed in facilities at Universidad Autónoma de Chile under the protocol BE-06-21, which was authorized by the Scientific Ethics Committee and Animal Bioethics, Universidad Autónoma de Chile. All experiments were conducted under the Animal Protection Law 20,380 guidelines of Chile and the *Guide for the Care and Use of Laboratory Animals* (8th Edition, National Academies Press, Washington, DC, USA).

### 4.2. Animal Model

The experiments were performed in 2-month-old male Sprague Dawley outbred rats. Animals were housed in individual cages in a temperature-controlled room on a 12 h light/12 h dark cycle, with food and water provided ad libitum. Since Sprague Dawley outbred rats do not show homogeneous voluntary alcohol consumption [61], we used i.p. ethanol administration to ensure the same daily dose in all animals. Ten two-month-old rats were administered ethanol 2 g/kg/day via i.p. (30% ethanol solution in saline) on Mondays, Wednesdays, and Fridays for four weeks. A group of 10 control rats was injected only with saline i.p. At the end of these four weeks, ethanol administration was terminated, and the ethanol-treated animals and controls were randomly separated into two groups each (n = 5). One ethanol group and one control group were administered micronized fenofibrate 50 mg/kg/day (Fibronil, Royal Pharma, Santiago, Chile)—resuspended in water—by esophageal gavage for 14 days. The other two groups (n = 5 each) were given only the vehicle, i.e., water by gavage. At the end of the treatments, the animals were deeply anesthetized with ketamine/xylazine (10:1 mg/kg of body weight, i.p.) and subjected to cardiac perfusion with saline until complete removal of blood from the lungs was evident. Then, the lungs were removed and washed with PBS buffer. The left lobules were fixed in 3.7% formalin buffered with PBS (pH 7.4) for histological analysis, and the right lobules were frozen immediately at −80 °C for molecular analyses (RT-qPCR and Western blot).

### 4.3. Histology

Paraffin-embedded sections, 5 μm thick, from the formalin-fixed lungs of the animals described in the previous point were deparaffinized according to the standard methods following a xylol and alcohol battery. Then, sections were stained with hematoxylin–eosin (H&E) or alcian blue (AB), following the standard procedures. To evaluate inflammation, we measure the proportion of infiltrated area using images with 100× magnification and using the following semi-quantitative score: 0: 0% infiltrated area; 1: <25% infiltrated area; 2: >25%–<50% infiltrated area; 3: >50%–<75% infiltrated area; 4: >75% infiltrated area. The total number of analyzed images was 5 per animal (25 per animal group). Sections stained with AB, according to the manufacturer’s instructions (DIAPATH, Martinengo, Italy), were analyzed to evaluate the quantity of mucus present in the airways. To this end, images of stained sections at 100× were analyzed to determine the bronchiolar epithelium covered by mucus. Images were opened in Image J software (version 1.53, NCBI, Bethesda, MD, USA) to select the epithelial area. Then, the red channel was chosen, and the threshold was determined to measure the stained region. The results were related to the total epithelial area and expressed as a percentage. The epithelial area was considered the region between the luminal surface of the airway and the basal membrane. According to the manufacturer’s instructions, collagen evaluations were performed using Masson Trichrome staining (DIAPATH, Martinengo, Italy). Collagen deposition was determined as the thickness by comparing the ratio of the measured stained area around the airway and the perimeter of the airway. All these analyses were conducted in 5 images per animal (25 images per animal group). All images were captured using an Olympus BX60 microscope and Image Pro Plus version 5.1.0 software (Media Cybernetics Inc., Rockville, MD, USA).

### 4.4. Determination of Inflammation Markers

Total RNA was extracted from 20 mg of fresh-frozen lung tissues extracted from the animals. Tissues were homogenized in the presence of 500 μL of RNA-Solv Reagent (Omega Biotek, Norcross, GA, USA), and the total RNA was purified according to the manufacturer’s indications. RNA samples were suspended in 50 μL of nuclease-free water and stored at −80 °C. Elimination of genomic DNA contaminants was made through incubation at 37 °C for 30 min of 2 μg of each RNA sample with 1 unit of DNAse (Thermo Fisher Scientific, Waltham, MA, USA) in a final volume of 10 μL. Enzyme activity was stopped by adding 2 μL of 25 mM EDTA and incubation at 65 °C for 10 min. Following this step, the RNA samples were heated at 70 °C for 5 min in the presence of 0.5 μg of random hexamer primers (Promega, Madison, WI, USA) at a final volume of 12.5 μL and rapidly placed on ice for 5 min. Then, nucleotides, buffer, RNAse inhibitor, and M-MLV reverse transcriptase were added to the reactions according to the manufacturer’s protocol (Promega, Madison, WI, USA), completing a final 25 μL volume. cDNA synthesis was performed according to the instructions of the M-MLV protocol (Promega, Madison, WI, USA). The resulting cDNAs were mixed with the addition of 50 μL of nuclease-free water and were used as templates for amplifying the inflammation markers Tnfα and Il6. The primers used are listed previously [62]. Amplification of these markers was performed using the Brilliant II SYBR QPCR Master Mix kit and the AriaMx qPCR equipment (Agilent, Santa Clara, CA, USA). All reactions were carried out in the following conditions: initial hot start at 95 °C for 12 min, followed by 45 cycles of 20 s at 95 °C, 20 s at 58 °C, and 20 s at 72 °C. Actin expression was used as an internal control. All results were expressed as fold changes in the level of mRNA of each gene normalized by the mRNA levels of actin and referenced to the control mock animal group using the 2^−ΔΔCt^ method [63].

### 4.5. Determination of Muc5ac Protein Levels

Protein extractions were made from 20 mg of fresh lung tissues from animals. These tissues were mixed with 500 μL of RIPA buffer (50 mM Tris at pH 7.4, 1% NP-40, 0.5% sodium deoxycholate, 150 mM NaCl, and one mM EDTA) supplemented by the addition of a complete protein inhibitor cocktail (Roche, Mannheim, Germany). Tissues were homogenized, and a final volume of 800 μL was completed with 300 µL of RIPA buffer and sodium dodecyl sulfate at a final concentration of 0.01%. Then, the samples were incubated in ice for 1 h. Final homogenization was carried out by passing the samples through a 21-gauge syringe three times. Proteins were obtained by centrifuging the samples for 15 min at 14,000× *g* at 4 °C two times and discarding the sedimented material. Proteins in supernatants were quantified using the Bradford Protein Kit (BioRad, Hercules, CA, USA) and stored at −80 °C. Thirty mg of each protein sample was used for Western blotting. To identify the mucin Muc5ac, proteins were separated in a 1.5% agarose gel prepared in TAE buffer and supplemented with sodium dodecyl sulfate at a final concentration of 0.1%. Electrophoresis was stopped when the running front reached 4 cm from the wells. Then, the gel was incubated for 5 min in the transference buffer while the PVDF membrane was activated according to the manufacturer’s instructions (Amersham Biosciences, Amersham, UK). Transference of the proteins was carried out in a Mini Trans-Blot Electrophoretic Transfer Cell (BioRad, Hercules, CA, USA) for one hour at 350 mA of constant current. Red Ponceau staining was performed as the internal control. Then, membranes were washed with TBS buffer and blocked using 5% non-fat milk prepared in TBS. The membranes were blocked for 2 h at room temperature. The blocking solution was discarded, and the membranes were washed with TBS. After washing, the membranes were probed with anti-Muc5ac (sc-21701, Santa Cruz Biotechnology, Dallas, TX, USA) overnight at 4 °C at dilutions of 1:2000 in 1% blocking solution diluted with TBS. Then, the membranes were washed by adding TBS supplemented with Tween-20 (T-TBS) at a 0.05% final concentration. After three washes of 10 min, the membranes were incubated for 1 h with an anti-mouse secondary antibody conjugated to HRP activity (sc-516102, Santa Cruz Biotechnology, Dallas, TX, USA) at 1:15,000 dilution in T-TBS. Then, the membranes were washed three times for 10 min using T-TBS. Protein complexes were visualized after incubation of the membranes with the SuperSignal West PICO Plus kit (Thermo Fisher Scientific, Waltham, MA, USA). Quantification of each protein band was made using Image J version 1.53 software. All results were expressed relative to the internal control actin and referenced to the control mock animal group.

### 4.6. Statistics

All statistical analyses were conducted using GraphPad Prism 10 (GraphPad Software Inc., San Diego, CA, USA). The results of the presented experiments are presented as mean ± standard deviation (SD). Each animal group consisted of 5 animals. The Shapiro–Wilk test was performed to determine the distribution of data. Data from 5 images (25 images in total per animal group) was grouped in a table. Differences between the animal groups were performed through a two-way ANOVA followed by Tukey´s multiple comparison test. In all the results, significance was defined as *p* < 0.05. All statistic data are presented as Appendix A.

## Figures and Tables

**Figure 1 ijms-25-12814-f001:**
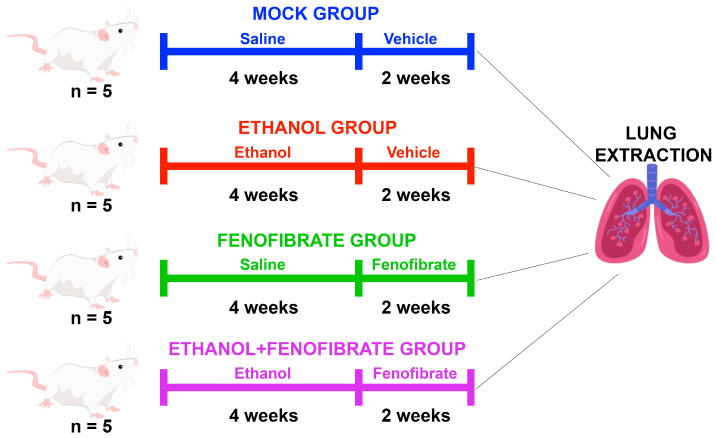
Experimental design. Animal model scheme of the experimental design used in the presented work. Animals were separated into four groups (n = 5): mock, ethanol, fenofibrate, and ethanol + fenofibrate. The ethanol injections (2 g/Kg) or water were applied during the first four weeks of the experiment 3 times per week. Then, a resting time of two weeks was applied to the animals, where fenofibrate (50 mg/Kg) or the vehicle was used as a treatment and delivered by gavage 3 times per week.

**Figure 2 ijms-25-12814-f002:**
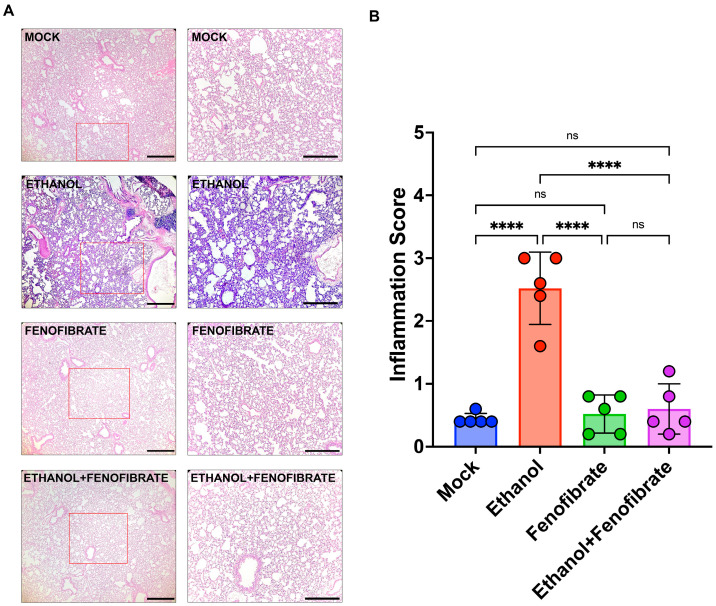
Fenofibrate reduces inflammation induced by ethanol. (**A**) Hematoxylin–eosin (H/E) stainings of lung sections from animals of all the experimental groups were examined by microscopy under 40× (left) and 100× (right) total magnification to identify the presence of inflammatory infiltrates. The images in the right column represent the magnification of the red square indicated in the pictures in the left column. The left scale bar indicates 500 μm, and the right scale bar indicates 100 μm. (**B**) Quantifications of stained sections were performed according to a scoring system described in Section 2. Data were analyzed using a two-way ANOVA test and expressed as mean ± SD. ns = non-significant; **** = *p* < 0.0001.

**Figure 3 ijms-25-12814-f003:**
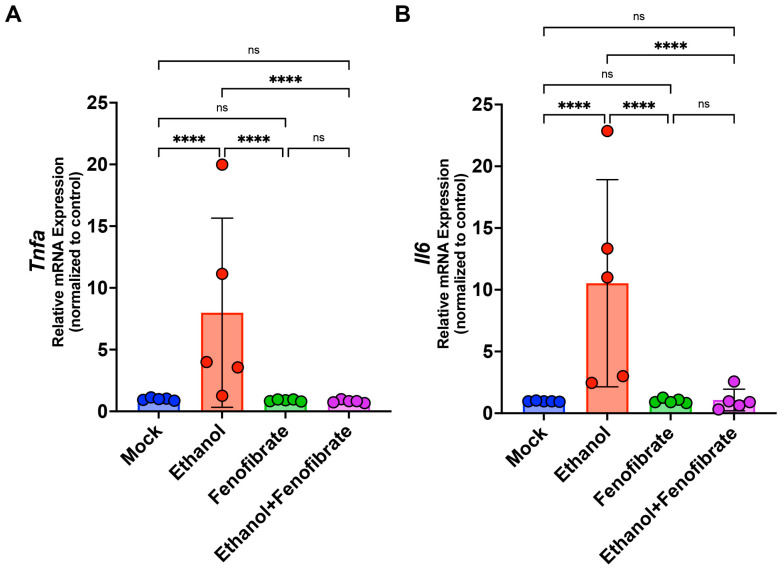
Fenofibrate reduces pro-inflammatory markers in animals injected with ethanol. Quantifying mRNA levels of Tnfα (**A**) and Il6 (**B**) was performed using qPCR. Data were analyzed using a two-way ANOVA test and expressed as mean ± SD. ns = non-significant; **** = *p* < 0.0001.

**Figure 4 ijms-25-12814-f004:**
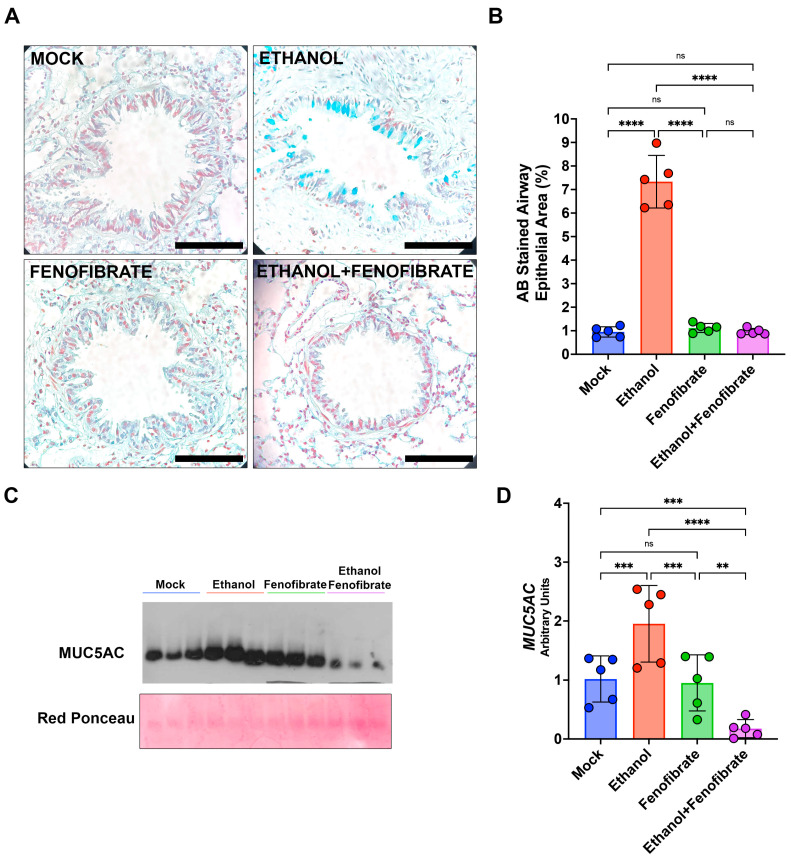
Fenofibrate reduces mucus in the airways of animals injected with ethanol. (**A**) Alcian blue (AB) stainings of lung sections from animals of all the experimental groups were examined by microcopy under 400× magnification to identify the presence of mucus in the airways. The bar indicates 100 μm. (**B**) Quantifications of the AB-stained area were performed according to the description presented in Section 2. (**C**) Western blot of MUC5AC showing three representative animals of each experimental group. Red Ponceau staining is shown as internal control. (**D**) Quantification of protein levels of MUC5AC in all animal groups. Data were analyzed using a two-way ANOVA test and expressed as mean ± SD. ns = non-significant; ** = *p* < 0.01; *** = *p* < 0.001, **** = *p* < 0.0001.

**Figure 5 ijms-25-12814-f005:**
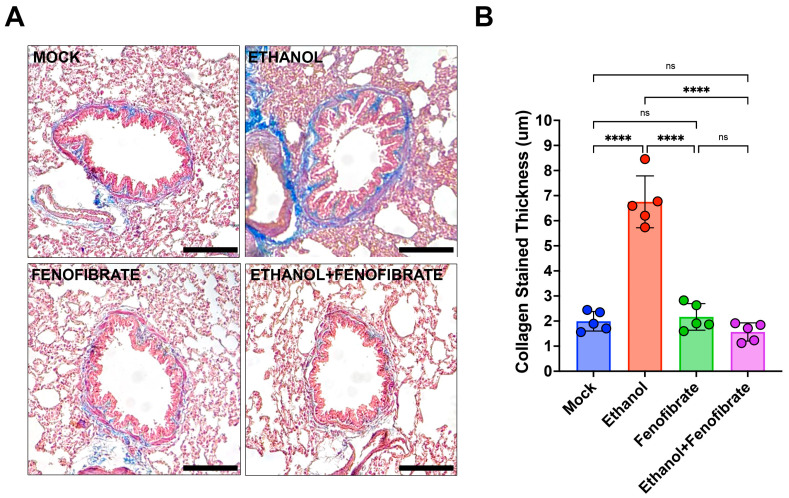
Fenofibrate reduces collagen deposition in the airways of animals injected with ethanol. (**A**) Masson Trichrome staining of lung sections from animals of all the experimental groups was examined by microcopy under 100× magnification to identify the presence of collagen deposition around the airways. The bar indicates 100 μm. (**B**) The collagen-stained area was quantified according to the description presented in Section 2. Data were analyzed using a two-way ANOVA test and expressed as mean ± SD. ns = non-significant; **** = *p* < 0.0001.

## Data Availability

The data presented in this study are available upon request from the corresponding authors.

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
