# Peer review of "Reduction of Alcohol-Dependent Lung Pathological Features in Rats Treated with Fenofibrate"

_ijms, 2024, doi:10.3390/ijms252312814_

Round 1

Reviewer 1 Report

Comments and Suggestions for Authors

Comments:-

This manuscript investigates the possibility of fibrate for treating inflammation induced by alcohol abuse. In my opinion, the overall strategy are organized systematically. Major suggestions have been raised for improve the quality of the manuscript.

1) Figure 1 should be revised as it is inappropriate for mixing figures and tables together. Maybe the table could be demostrated by schematic diagram.

2) 40X (left) and 100X (right) are not commonly used for microcopy images. Maybe clear scar bar is better.

3) “The treatment of fenofibrate to reduce neuroinflammation has been described in alcohol-drinking rats”. You should explain for the mechanism of the drug in reducing inflammation in the Introduction section.

4) It is suggested to review the comparison of current anti-inflammation drugs with fenofibrate in Introduction. Why not use common anti-inflammation drugs? Actually, fenofibrate has more reverse effect and has no anti-inflammatory indications.

Author Response

1) Figure 1 should be revised as it is inappropriate for mixing figures and tables together. Maybe the table could be demostrated by schematic diagram.

RESPONSE: Thank you for the suggestion. We have modified Figure 1 according to your comments. We have eliminated the table of the figure and added the information on the dose of alcohol/fenofibrate in the animal model scheme of the figure.

2) 40X (left) and 100X (right) are not commonly used for microcopy images. Maybe clear scar bar is better.

RESPONSE: Thank you for your comment. We have published other works using the same magnifications elsewhere. However, we have improved the scale bar in Figure 2 to a better understanding of the presented histology.

3) “The treatment of fenofibrate to reduce neuroinflammation has been described in alcohol-drinking rats”. You should explain for the mechanism of the drug in reducing inflammation in the Introduction section.

RESPONSE: Thank you for your comments. We have added an additional text explaining this point in the introduction section. Specifically, focusing in the role of NFkB pathway described in 10.1016/j.neuropharm.2017.11.003.

4) It is suggested to review the comparison of current anti-inflammation drugs with fenofibrate in Introduction. Why not use common anti-inflammation drugs? Actually, fenofibrate has more reverse effect and has no anti-inflammatory indications.

RESPONSE: Thanks for this comment, you are right. We have added a paragraph in the discussion section indicating that fenofibrate has anti-inflammatory properties through the inhibition of NF-kB activity, which impacts innate and adaptive immune responses, while classical non-steroidal anti-inflammatory drugs (NSAIDs) have a different mechanism of action through the inhibition of cyclooxygenases inhibiting prostaglandins production. In addition, the election of fenofibrate was based on our previous work, where we demonstrated its potent effects in reducing ethanol-induced neuroinflammation.

Reviewer 2 Report

Comments and Suggestions for Authors

A review of : “Reduction of Alcohol-Dependent Lung Pathological Features in 2 Rats Treated with Fenofibrate” by Rojas and colleagues.

In this paper the authors explored the effects of repeated alcohol IP administration on lung pathology and the potential effects of fenofibrate treatment, a PPAR alpha agonist, on alcohol-induced pathological measurements. Overall, the idea of using fenofibrate is novel. However the paper has some major problems that must be addressed.

Major comments:

1)        Data analysis of histological data in figures 2,4,5 was done with considering each image or slice as a sample. However, there were only 5 mice per group. Each mouse is one biological replicate. It is misleading to refer to each slice as an independent measurement as slices from the same mouse are not independent. The authors should either sum or average the results from each slice within each mouse and then analyze the data with n=5 per group. Alternatively, they can handle different slices from the same mouse replicate if they use nested ANOVAs.

2)        Statistical analysis to detect significant differences between groups, the authors used One-way ANOVA for all their experiments. Since there are two major factors in their experiments: 1- alcohol/mock and 2- fenofibrate/water. The appropriate analysis should be two-way ANOVA followed by post-hocs.

3)        Fenofibrate is a PPAR alpha agonist that negatively regulates NfKB, as the authors mentioned in the discussion. It is very surprising that NfKB was not quantified by qPCR or other measures, which seems very important. It would show that the drug worked as intended and is an additional very relevant proinflammatory gene. I suggest quantifying NfKB in these samples.

4)        Western blot (fig4 and supp fig). Mucin5AC was measured on one membrane after running protein samples on agarose gel and actin on another membrane after running protein samples on Polyacrylamide gel. The use of agarose gel is because Mucins are very large proteins (a few hundred kilo Daltons). However, it is impossible to normalize protein levels between the two membranes. The actin (or any other housekeeping protein) should control for variation in loading and protein transfer efficiency (gel to the membrane), both are not relevant when running the samples on two membranes and therefore the quantification is wrong and misleading (also n=3 /group is very low). The authors can fix it with one of several ways: 1- Using a large housekeeping protein for control (on the same membrane) although this might be hard to find such a protein. 2- quantifying total protein on the membrane (quantification of Ponceau red signal for example), 3- quantifying Mucin5AC by ELISA or 5- quantifying Mucin5AC by immunohistochemistry. I think 5 would be the easiest as this team showed they are familiar with lung histology.

Minor comments:

11)        Section 2.2: “However, a few pieces of evidence have shown the relationship between 143 alcohol intake and mucus in the airways.”  I think “piece of evidence” is not a common phrase in English; please correct.

22)        In the abstract text, it is not immediately clear when the author describes common previous knowledge and when they start to describe their own findings.

Comments on the Quality of English Language

English is OK, minor edits would help as detailed in my comments to the authors.

Author Response

Major comments:

1)        Data analysis of histological data in figures 2,4,5 was done with considering each image or slice as a sample. However, there were only 5 mice per group. Each mouse is one biological replicate. It is misleading to refer to each slice as an independent measurement as slices from the same mouse are not independent. The authors should either sum or average the results from each slice within each mouse and then analyze the data with n=5 per group. Alternatively, they can handle different slices from the same mouse replicate if they use nested ANOVAs.

RESPONSE: Thank you for your comment. You are completely right. In this new version of the manuscript, we have generated a new analysis with data performing a two-way ANOVA to incorporate the measurements of each animal in all the results. The description of the measurements was added to the methods section. All the graphics presented in the new figures have been adjusted according to these new analyses. Thank you for detecting this mistake in our results.

2)        Statistical analysis to detect significant differences between groups, the authors used One-way ANOVA for all their experiments. Since there are two major factors in their experiments: 1- alcohol/mock and 2- fenofibrate/water. The appropriate analysis should be two-way ANOVA followed by post-hocs.

RESPONSE: Thank you for your suggestion. As we commented in the previous paragraph, we have improved our statistical analysis by performing two-way ANOVA in all our determinations. All the figures were improved incorporating these new analyses.

3)        Fenofibrate is a PPAR alpha agonist that negatively regulates NfKB, as the authors mentioned in the discussion. It is very surprising that NfKB was not quantified by qPCR or other measures, which seems very important. It would show that the drug worked as intended and is an additional very relevant proinflammatory gene. I suggest quantifying NfKB in these samples.

RESPONSE: Thank you for your comment. We have added a paragraph specifically explaining the mechanism by which fenofibrate (activating PPARα) leads to the inhibition of NF-kB activity. As explained in the new version of the manuscript, this mechanism is not related to the decrease in NF-kB expression, but to an inhibition of its ability to enter the nucleus from the cytoplasm. In this context, quantification of the mRNA or protein corresponding to NF-kB would not be useful. Moreover, to identify the modulation of NF-kB activity, we need to evaluate the abundance of p65 sub-unit in cell compartments such as cytoplasm and nuclei; however, this would lead us to additional experiments, such as nuclear extracts evaluations or EMSAs. In addition, the detection of cytokine TNFα shows that this pathway is modulated in the context of our experiments.

4)        Western blot (fig4 and supp fig). Mucin5AC was measured on one membrane after running protein samples on agarose gel and actin on another membrane after running protein samples on Polyacrylamide gel. The use of agarose gel is because Mucins are very large proteins (a few hundred kilo Daltons). However, it is impossible to normalize protein levels between the two membranes. The actin (or any other housekeeping protein) should control for variation in loading and protein transfer efficiency (gel to the membrane), both are not relevant when running the samples on two membranes and therefore the quantification is wrong and misleading (also n=3 /group is very low). The authors can fix it with one of several ways: 1- Using a large housekeeping protein for control (on the same membrane) although this might be hard to find such a protein. 2- quantifying total protein on the membrane (quantification of Ponceau red signal for example), 3- quantifying Mucin5AC by ELISA or 5- quantifying Mucin5AC by immunohistochemistry. I think 5 would be the easiest as this team showed they are familiar with lung histology.

RESPONSE: Thank you for your suggestion. Working with agarose is common when we are evaluating high molecular weight proteins such as mucin. However, evaluating other proteins, such as actin, is more complicated because their migration is not adjusted to the protein ladder. So, it is complicated to evaluate the time of the gel running. In addition, we have determined that the quality of the detection of actin is not the best in the western blot. However, we have replaced the image of the western blot with a new experiment showing the detection of actin evaluated previously in an agarose matrix. With respect to the number of evaluations, we have added the determination of all the animals, and the other experiments are attached as non-published supplementary material.

Minor comments:

11)        Section 2.2: “However, a few pieces of evidence have shown the relationship between 143 alcohol intake and mucus in the airways.”  I think “piece of evidence” is not a common phrase in English; please correct.

RESPONSE: Thank you for your comment. We have corrected this lane. In addition, we have edited all the text again to identify other grammatical mistakes.

22)        In the abstract text, it is not immediately clear when the author describes common previous knowledge and when they start to describe their own findings.

RESPONSE: Thank you for your comment. You are right. We have improved the abstract according to your suggestion.

Round 2

Reviewer 1 Report

Comments and Suggestions for Authors

I recommend accepting the paper.

Author Response

Thank you for your comments.

Reviewer 2 Report

Comments and Suggestions for Authors

A review of the revised version of : “Reduction of Alcohol-Dependent Lung Pathological Features in 2 Rats Treated with Fenofibrate” by Rojas and colleagues.

In this revised version, the author answered most of my comments and provided a corrected statistical analysis. Therefore, this version of the manuscript is much better. However, the authors’ response to the criticism of the western blot of Mucin5ac is, unfortunately, not adequate. The authors provide an image of a previous blot of actin on agarose gel. This does not control in any way for variation in sample loading and protein transfer efficiency within the same membrane. Therefore any quantification of Mucin5ac based on this experiment is wrong. This can be fixed as suggested in my first round of comments with one of several approaches (see below), otherwise any claims about Mucin5ac in this study are premature. The potential approaches are: 1- Using a large housekeeping protein for control (on the same membrane) although this might be hard to find such a protein. 2- quantifying total protein on the membrane (quantification of Ponceau red signal for example), 3- quantifying Mucin5AC by ELISA or 4- quantifying Mucin5AC by immunohistochemistry.

Minor comments:

1)        For the ANOVAs, please provide also the F values and degrees of freedom for each main effect. Also, no statistical interactions within the ANOVAs (or the lack of) were reported; please add that as well.

Author Response

In this revised version, the author answered most of my comments and provided a corrected statistical analysis. Therefore, this version of the manuscript is much better. However, the authors’ response to the criticism of the western blot of Mucin5ac is, unfortunately, not adequate. The authors provide an image of a previous blot of actin on agarose gel. This does not control in any way for variation in sample loading and protein transfer efficiency within the same membrane. Therefore any quantification of Mucin5ac based on this experiment is wrong. This can be fixed as suggested in my first round of comments with one of several approaches (see below), otherwise any claims about Mucin5ac in this study are premature. The potential approaches are: 1- Using a large housekeeping protein for control (on the same membrane) although this might be hard to find such a protein. 2- quantifying total protein on the membrane (quantification of Ponceau red signal for example), 3- quantifying Mucin5AC by ELISA or 4- quantifying Mucin5AC by immunohistochemistry. 

RESPONSE: Thank you for your comment. We thought that red ponceau staining was not completely adequate for quantification evaluations. That is the reason why we attached a new experiment determining actin levels using an agarose gel in the last version of our manuscript. However, in this new manuscript version, we have added the images associated with the Red Ponceau staining performed on our membranes before Muc5ac detection. We changed the plot of the figure 4 according to this new quantification.

Minor comments:

1)        For the ANOVAs, please also provide the F values and degrees of freedom for each main effect. Also, no statistical interactions within the ANOVAs (or the lack of) were reported; please add that as well.

RESPONSE: Thank you for your comments on this point. We have attached a supplementary material file with all the statistical information exported from the software GraphPad associated with all the graphics presented in the manuscript. We hope this file contains all the required information.

Round 3

Reviewer 2 Report

Comments and Suggestions for Authors

The authors answered all my comments.

However, the supplementary materials with the statistics was not provided to me to review.